# A Comparison of Cervical and Thoracolumbar Fractures Associated with Diffuse Idiopathic Skeletal Hyperostosis—A Nationwide Multicenter Study

**DOI:** 10.3390/jcm9010208

**Published:** 2020-01-12

**Authors:** Hiroyuki Katoh, Eijiro Okada, Toshitaka Yoshii, Tsuyoshi Yamada, Kei Watanabe, Keiichi Katsumi, Akihiko Hiyama, Yukihiro Nakagawa, Motohiro Okada, Teruaki Endo, Yasuyuki Shiraishi, Kazuhiro Takeuchi, Shunji Matsunaga, Keishi Maruo, Kenichiro Sakai, Sho Kobayashi, Tetsuro Ohba, Kanichiro Wada, Junichi Ohya, Kanji Mori, Mikito Tsushima, Hirosuke Nishimura, Takashi Tsuji, Kota Watanabe, Morio Matsumoto, Atsushi Okawa, Masahiko Watanabe

**Affiliations:** 1Department of Orthopaedic Surgery, Surgical Science, Tokai University School of Medicine, Kanagawa 259-1193, Japan; ha084442@tsc.u-tokai.ac.jp (A.H.); masahiko@is.icc.u-tokai.ac.jp (M.W.); 2Japanese Organization of the Study for Ossification of Spinal Ligament (JOSL), Tokyo 113-8519, Japan; eijiro888@gmail.com (E.O.); kkatsu_os@yahoo.co.jp (K.K.); yukihiro19670916@gmail.com (Y.N.); motohiro2doctor@yahoo.co.jp (M.O.); teruaki@jichi.ac.jp (T.E.); r0728ys@jichi.ac.jp (Y.S.); takeuchi@okayamamc.jp (K.T.); shunji@imakiire.or.jp (S.M.); kenitiro1122@gmail.com (K.S.); dr.shokobayashi@gmail.com (S.K.); tooba@yamanashi.ac.jp (T.O.); jun.ohya@gmail.com (J.O.); meikeihan@hotmail.com (M.T.); hirosuke819@hotmail.com (H.N.); tsuji9@gmail.com (T.T.); morio@a5.keio.jp (M.M.); okawa.orth@tmd.ac.jp (A.O.); 3Department of Orthopaedic Surgery, Keio University School of Medicine, Tokyo 160-8582, Japan; 4Department of Orthopaedic Surgery, Tokyo Medical and Dental University, Tokyo 113-8519, Japan; 5Department of Orthopaedic Surgery, Niigata University, Niigata 951-8510, Japan; 6Department of Orthopaedic Surgery, Wakayama Medical University Kihoku Hospital, Wakayama 649-7113, Japan; 7Department of Orthopaedic Surgery, Wakayama Medical University, Wakayama 641-8509, Japan; 8Department of Orthopaedic Surgery, Jichi Medical University, Tochigi 329-0498, Japan; 9Department of Orthopaedic Surgery, National Hospital Organization Okayama Medical Center, Okayama 701-1192, Japan; 10Department of Orthopaedic Surgery, Imakiire General Hospital, Kagoshima 892-8502, Japan; 11Department of Orthopaedic Surgery, Hyogo College of Medicine, Hyogo 663-8501, Japan; 12Department of Orthopaedic Surgery, Saiseikai Kawaguchi General Hospital, Saitama 332-8558, Japan; 13Department of Orthopaedic Surgery, Hamamatsu University School of Medicine, Shizuoka 432-8580, Japan; 14Department of Orthopaedic Surgery, Yamanashi University, Yamanashi 409-3898, Japan; 15Department of Orthopaedic Surgery, Hirosaki University Graduate School of Medicine, Aomori 036-8562, Japan; 16Department of Orthopaedic Surgery, The University of Tokyo, Tokyo 113-8655, Japan; 17Department of Orthopaedic Surgery, Shiga University of Medical Science, Shiga 520-2192, Japan; 18Department of Orthopaedic Surgery, Chubu Rosai Hospital, Aichi 455-8530, Japan; 19Department of Orthopaedic Surgery, Tokyo Medical University, Tokyo 160-8402, Japan

**Keywords:** diffuse idiopathic hyperostosis, spine trauma, ankylosis, trauma

## Abstract

In diffuse idiopathic hyperostosis (DISH), the ankylosed spine becomes susceptible to spinal fractures and spinal cord injuries due to the long lever arms of the fractured segments that make the fracture extremely unstable. The aim of this retrospective multicenter study was to examine the differences in DISH-affected spine fractures according to fracture level. The data of 285 cases with fractures of DISH-ankylosed segments diagnosed through computed tomography (CT) imaging were studied and the characteristics of 84 cases with cervical fractures were compared to 201 cases with thoracolumbar fractures. Examination of the CT images revealed that cervical fracture cases were associated with ossification of the posterior longitudinal ligament and had fractures at the intervertebral disc level, while thoracolumbar fracture cases were associated with ankylosing of the posterior elements and had fractures at the vertebral body. Neurologically, cervical fracture cases had a higher ratio of spinal cord injury leading to higher mortality, while thoracolumbar fracture cases had lower rates of initial spinal cord injury. However, a subset of thoracolumbar fracture cases suffered from a delay in diagnosis that led to higher rates of delayed neurological deterioration. Some of these thoracolumbar fracture cases had no apparent injury episode but experienced severe neurological deterioration. The information provided by this study will hopefully aid in the education of patients with DISH and raise the awareness of clinicians to potential pitfalls in the assessment of DISH trauma patients.

## 1. Introduction

Diffuse idiopathic hyperostosis (DISH) is a condition characterized by a tendency toward ossification of ligaments, tendons, and enthesial insertions, with manifestations being most prominent in the spine. The condition was first reported as senile vertebral ankylosing hyperostosis in 1950 by Forestier and Rotes-Querol in a report of nine patients suffering from spinal rigidity with exuberant osteophytes on plain radiographs [1]. Reporting on the extraspinal manifestations of the condition, Resnick later proposed the term DISH and introduced the radiological criteria of the condition [2]. Interestingly, research shows that DISH affects not only humans but also many mammals and even dinosaurs [3,4,5], suggesting that DISH is not a disease, but rather an age-dependent condition that is seen across many species. 

DISH is often an asymptomatic condition that is incidentally discovered on imaging studies, and many people go about their daily lives oblivious to the insidious ossification and ankylosing that is progressing in their spine. As the progression of vertebral fusion becomes extensive, the reduction of spinal column mobility can limit motion and cause pain in the neck to lower back, and an ossified mass in the anterior cervical spine can cause dysphagia. An important characteristic of an ankylosed spine is its increased susceptibility to spinal fractures and spinal cord injuries due to the long lever arms of the fractured segments that make the fracture extremely unstable. 

Patients with DISH-associated spinal fractures can often experience a delay in diagnosis. The combination of pre-existing pain and the mild nature of the injury episode can lead the patients or their physicians to underestimate the severity of injury. Furthermore, some patients do not initially experience any neurological symptoms until a specific point when they suffer a sudden deterioration, often with severe and permanent sequelae in the form of a spinal cord injury (SCI). An increasing number of elderly patients suffer traumatic spinal cord injuries due to falls from a standing/sitting height [6,7]. Many of these cases are diagnosed as cervical SCI without radiographic abnormalities (SCIWORA), but SCI in DISH-ankylosed spines are increasing. With the long lever arms of the fracture segments making the fracture site extremely unstable and at high risk for SCI, surgical fixation has been documented to be superior to conservative treatment.

In order to examine the characteristics of spinal fractures in DISH-ankylosed segments, a retrospective multicenter study was conducted under the initiative of the Japanese Organization for the Study for Ossification of Spinal Ligament (JOSL). Patients who were definitively diagnosed by CT as having a fracture through the DISH-ankylosed spinal segment were retrospectively collected from JOSL-affiliated institutions. With the initial report of the general findings of this study recently published [8], the aim of this follow-up report is to examine the differences in DISH-affected spine fractures according to fracture level, particularly between the cervical and thoracolumbar spine, and compare the timing of diagnosis, neurological status, and radiological characteristics of injuries.

## 2. Methods

### 2.1. Study Design and Subject Criteria

This is a retrospective multicenter case-series study undertaken by JOSL with ethical approval obtained from the institutional review board of each participating institution. Each institution was directed to register cases that met the following criteria: (1) spine trauma patients treated during the period of January 2005 to December 2015, (2) diagnosed with DISH according to the criteria set by Resnick and Niwayama (flowing ossification along the anterolateral aspect of at least four contiguous vertebral bodies [2]), (3) suffered a spinal fracture through the DISH-ankylosed spinal segment, (4) definitively diagnosed by multiplanar reconstructed CT imaging, and (5) had no history of prior spine surgery. From a total of 307 cases that were registered from 18 participating institutions, 285 cases were the subject of this study after 22 cases were excluded due to the following reasons: 15 cases because the fracture was in a level not fused by DISH, 6 cases because CT imaging was not available, and 1 case because the fracture was not apparent on CT images. 

### 2.2. Data Collection

The following data for each case were evaluated and registered by an author at each participating institution. The cases were categorized as either a cervical or a thoracolumbar fracture case according to fracture level, and the accumulated data were compared between the two groups.

#### 2.2.1. Demographics and Cause of Injury

Demographic data collected for each case were age, gender, height, weight, and past medical history. While associated diabetes mellitus was specifically enquired for each case, all other comorbidities and past medical history were provided in a free form; therefore, the list of past medical history may not be complete for all cases. The cause of injury was catalogued as either a fall from a standing/sitting position, a fall from a height, a traffic accident, or other, in which case the physician was asked to register the cause of injury in a free form.

#### 2.2.2. Radiological Characteristics

The CT images were analyzed, and the radiological features were catalogued for each case by the primary physician. All fracture levels within the DISH-ankylosed segments were registered, but for the 25 cases that suffered more than one fractured vertebra, the fracture site that was judged to be most severe, i.e., the level responsible for neurological symptoms or with the greatest fracture displacement, was used to categorize the case as either cervical or thoracolumbar. The CT images were also examined for the presence of ossification of posterior longitudinal ligament (OPLL), ossification of ligamentum flavum (OLF), and ankylosing of posterior elements at the fracture site.

All cases had fractures of the anterior elements, i.e., the vertebral body, intervertebral disc space, or a fracture line that transected both elements. The anterior fracture line was catalogued as a vertebral body fracture if ≥ 50% of the fracture line through the anterior vertebral column runs through the vertebral body and intervertebral if ≥ 50% of the fracture line runs through the intervertebral disc [8]. The fracture pattern was also categorized as Type 1 to Type 4 according to the classification proposed by Caron [9]. Fractures of the posterior elements at the same level were also recorded. 

#### 2.2.3. Delay in Diagnosis and Neurological Status

A delayed diagnosis was defined as a fracture that was not diagnosed within 24 h of the injury; it was categorized as patient delay if the patient did not seek medical attention during that period, and doctor delay if the patient visited a hospital but was not initially diagnosed as a spinal fracture [10]. Neurological status immediately after injury and at follow-up were recorded according to the Frankel grade [11], and the physician was also asked if any neurological worsening was observed during the course of treatment.

### 2.3. Statistical Analyses 

All statistical analyses were performed on SPSS Statistics 25 (IBM Corp, New York, NY, United States), and a *p* value less than 0.05 was considered statistically significant. After verifying homogeneity of variances by Levine’s test for equality of variances, an independent-samples t-test was run to determine if there were differences in age and body mass index (BMI); data are expressed as the mean ± standard deviation. A chi-square test of homogeneity was run to determine if there were differences between cervical and thoracolumbar fracture groups for all nominal parameters. When significant differences were observed between groups, post hoc analyses involving pairwise comparisons with multiple z-tests of two proportions were performed with a Bonferroni correction, and statistical significance was set according to the number of pairwise comparisons performed (0.05 / number of pairwise comparisons). 

## 3. Results

### 3.1. Demographics and Cause of Injury

The 285 subjects of the study were comprised of 84 cases with cervical fractures and 201 cases with thoracolumbar fractures, and their demographic data are presented in Table 1. Reflecting the population that is usually affected by DISH, the subjects were predominantly elderly males with an average age of 75.1 years. The cervical group had a significantly higher proportion of male cases, but there was no difference in age between the two groups. Impaired glucose tolerance has been suggested to be involved in the development of DISH, and there was no statistical difference in the number of cases with diabetes between the two groups. With DISH affecting the elderly population, the effect of osteoporosis cannot be neglected, but there were only five cases that were being treated for osteoporosis before the fracture episode. Data concerning osteoporosis were requested for this study, but with only bone mineral density data collected from 21 cases, a comparison was not attempted. There was also no significant difference observed between the two groups regarding kidney or liver diseases, history of past fractures, or malignancies. 

A review of the cause of injury revealed that approximately half of the cases in both groups sustained injury through a minor trauma such as falling from a standing or sitting position, but there was no correlation between cause of injury and fracture level. Interestingly, the thoracolumbar group had a significantly higher number of cases that reported no apparent injury episode. 

### 3.2. Radiological Characteristics

The cumulative number of fractures per vertebral level is presented in Figure 1, showing a bimodal distribution. Analysis of the CT images revealed that cervical cases were associated with ossification of the posterior longitudinal ligament (OPLL) and thoracolumbar cases often had ankylosed posterior elements. A comparison of the fracture morphology revealed that cervical fractures occurred mainly at the intervertebral disc level, while thoracolumbar fractures occurred mainly at the vertebral body level (Table 2 and Figure 2). 

Accompanying ossification is often observed in the DISH-ankylosed spine, but the contrasting ossification pattern between cervical and thoracolumbar cases was not expected. Reflecting the higher association of OPLL with the cervical spine, approximately half of the cervical cases had OPLL of the fracture site, while OPLL was rare in thoracolumbar cases. What was often observed at the fracture site of thoracolumbar cases was fusion of the posterior elements, which was significantly higher than in cervical cases (Table 2). The relationship between ossification at the fracture site and age was examined, but no apparent correlation was observed.

The fracture morphology was also quite different between cervical and thoracolumbar fractures. When categorized as either intervertebral or vertebral body fracture by evaluating in which area the anteroposterior length of the fracture line was longer, 65.5% of cervical fractures were intervertebral while 75.6% of thoracolumbar fractures occurred at the vertebral body (Table 2 and Figure 2). Evaluation of fracture pattern using the classification proposed by Caron also confirmed that cervical fracture cases had significantly more fractures through the disc space while most thoracolumbar fracture cases had fractures at the vertebral body (Table 2). Possibly reflecting the high percentage of posterior column ankylosing, the thoracolumbar fracture group had a significantly higher association with fractures of the posterior elements. The relationship between cause of injury and fracture morphology was also evaluated, but no statistically significant correlation was observed.

### 3.3. Delay in Diagnosis and Neurological Status

The delayed diagnosis of a fracture within a DISH-ankylosed segment can have devastating neurological consequences. A comparison of diagnosis delay between cervical and thoracolumbar cases revealed that thoracolumbar fracture cases had significantly higher rates of delay. This may in part be due to the difference in neurological status of the two groups, because the cervical fracture group had significantly greater number of cases with initial SCI after injury. However, a troubling number of thoracolumbar cases suffered from delayed deterioration of neurological status, often causing neurologically intact cases to become paralyzed from SCI. 

Of the 285 cases enrolled in this study, 114 cases (40.0%) experienced a delay before being diagnosed as having a fracture of the ankylosed spine segment afflicted with DISH. With the precipitating injury being minor in half of the cases, the patient’s delay is understandable because they did not consider their fall severe enough to cause a serious problem; but the doctor’s delay is disturbing. The thoracolumbar fracture group had significantly higher rates of diagnosis delay, due to both the patient and their physician (Figure 3a). It is noteworthy that neurological worsening was significantly higher in the thoracolumbar cases with delayed diagnosis (*p* < 0.01, Figure 3b) and was especially high (45.6%) in thoracolumbar cases with doctor’s delay. 

The initial and final neurological status is presented in Figure 4a, revealing that 76.2% of the cervical group suffered SCI immediately after injury, which was significantly higher than the thoracolumbar group (34.3%, *p* < 0.01). Many of the patients in the thoracolumbar group did not have neurological symptoms initially, but a significantly greater number of cases in the thoracolumbar fracture group suffered from delayed neurological deterioration compared to cervical cases (*p* < 0.05, Figure 4a–c). 

When the entire cohort was analyzed, the ratio of SCI was significantly higher in the OPLL-positive group (*p* < 0.01). However, further analysis of the association of OPLL and SCI in the cervical fracture group did not reveal any significant difference (*p* = 0.091), suggesting that the high association of OPLL in the cervical group was a confounding factor that led to the statistical association of SCI with OPLL.

### 3.4. Thoracolumbar Cases with Neurological Deterioration

Initially, 132 cases in the thoracolumbar fracture group did not have SCI (Frankel grade E), but 31 cases later suffered a delayed conversion to SCI (Figure 4a). The data from 47 thoracolumbar fracture cases that experienced delayed neurological deterioration (Figure 5) suggest that the mild injury episode and lack of neurological symptoms misled the patients and physicians to believe that the injury was not severe. 

The number of deteriorated cases in this group is greater than the number of thoracolumbar cases shown to have suffered deterioration in Figure 4b, because there were cases that experienced minor deterioration without changes in Frankel grading. The cause of injury in this group was similar to the entire study population, with more than half of the cases suffering injury through a minor episode, such as a fall from a sitting or standing position, but there were also seven cases that had no recollection of any precipitating trauma. A delay in diagnosis occurred in 76.6% of the cases in this group—of which, three quarters were doctor’s delay. When the reasons for delay were enquired, mild pain, missed diagnosis by their physician, and initial imaging with only plain radiographs were equally listed. It is noteworthy that there was a large variation in the timing of neurological deterioration. With 22 cases reporting paralysis later than 1 week after injury, it is admittedly difficult to suspect a fracture.

### 3.5. Treatment and Complications

Table 3 outlines the treatment, complications, and mortality data of the cervical and thoracolumbar fracture groups. Most cases in both groups were treated with surgery, with posterior fusion being the surgical method of choice in approximately 80% of the cases. Reflecting the advanced years of the patients, the complication rates were high with 36.9% of cervical cases and 27.4% of thoracolumbar fracture cases experiencing at least one complication. Respiratory complications were the most frequent complication experienced in the cervical group, which was often associated with the loss or decrease in respiratory function due to SCI, requiring ventilator assistance. Dysphagia requiring a gastrostomy was reported in three cases—two of which had anterior fusion surgery. In the thoracolumbar group, instrument-related complications were the most frequent. Death within 6 months of surgery occurred in 15.5% of cervical cases, which was significantly higher compared to thoracolumbar cases (*p* < 0.05).

## 4. Discussion

This study is a multicenter retrospective study of cervical and thoracolumbar fractures associated with DISH, aimed to assess and compare the timing of diagnosis, neurological status and radiological characteristics of injuries. As the largest study of fractures in DISH cases, Okada reported the general findings of the study elsewhere [8], and this report further analyzes the differences according to fracture level. Since fractures of the ankylosed DISH segments were distributed in a bimodal distribution centered at the lower cervical level and the thoracolumbar junction, this study compared the characteristics between cervical fractures and thoracolumbar fractures. The differences were more pronounced than expected, with cervical fracture cases suffering from a higher degree of SCI and mortality, while the initial neurological condition of thoracolumbar fracture cases was considerably less severe. However, the higher rates of delayed diagnosis and neurological deterioration in thoracolumbar fracture cases demonstrates that the mild symptoms can sometimes lead to a false sense of security both for the patient and physician, leaving some patients vulnerable to a sudden deterioration with severe and permanent sequelae in the form of a SCI. 

In this study, fractures in the DISH-ankylosed spine were heavily oriented toward thoracolumbar cases, which is different from the literature that reports more cervical than thoracolumbar fractures in cases with DISH: the percentages of cervical and thoracolumbar fractures were reported as 62.8% and 37.2 by Westerveld in 2009 [10], 52.5% and 47.5% by Westerveld in 2014 [12], 61.5% and 57.7% by Teunissen [13], and 50% each by Caron [9]. While the effect of race in DISH-affected levels cannot be excluded, the greater number of fractures in the thoracolumbar region seems natural because the highest prevalence of DISH has been reported to be at the T8–T10 levels in Japanese patients [14,15]. High prevalence of cervical OPLL may have also affected data collection, because some DISH+OPLL cases may have been catalogued only as cervical spine fracture with OPLL and missed by the retrospective screening for DISH-associated fractures.

### 4.1. Prevalence of DISH-Associated Spine Fractures

There have been a number of studies concerning the complications and outcomes of the ankylosed spine, but most articles evaluated and compared ankylosing spondylitis (AS) and DISH [9,10,12,16]. While the prevalence of AS per 10,000 has been reported to be approximately 23.8 in Europe, 16.7 in Asia, 31.9 in North America, 10.2 in Latin America, and 7.4 in Africa [17], the prevalence of AS in Japan is significantly lower at 0.65 [18] and thus fractures in AS-ankylosed spines are rare. The rapidly aging population of Japan makes the prevalence of DISH much higher, but there are varying reports on the prevalence of DISH in the Japanese population. Kagotani studied the prevalence of DISH in a population-based cohort study of 1647 cases with whole-spine radiographs and found DISH to be present in 10.8% of cases [19]. However, the sensitivity of plain radiographs to discern early stage DISH is questionable, and many studies have investigated DISH prevalence using CT imaging. Mori studied chest CT scans of 3013 Japanese patients and reported the prevalence of thoracic DISH as 8.7% [20]. Fujimori evaluated 1500 Japanese patients who underwent positron emission tomography and CT and reported a 12% prevalence of DISH [21]. Hiyama evaluated the prevalence of DISH in 1479 trauma patients who were screened for injury using head-to-pelvis CT scans and reported a 19.5% prevalence [14]. While the true incidence of DISH among the Japanese population remains unknown, spine surgeons in Japan are increasingly encountering spinal fractures in patients with DISH. With aging of the population progressing worldwide, this study is performed in the hopes of adding to the literature on fractures in the DISH-afflicted spine.

### 4.2. Fracture Characteristics

With DISH afflicting the elderly population, fractures at adjacent levels to the ankylosed segments may be affected by osteoporosis. However, DISH does not cause osteoporosis of the vertebra [22] and so fractures in the ankylosed DISH spine are believed to occur at the point of least resistance. In this study, a majority of fractures in the cervical spine were through the intervertebral disc space, while most fractures in the thoracolumbar spine went through the vertebral body. This is in line with the findings of Caron who reported intervertebral fractures in 51% of cervical cases (Types I and IV) [9] as well as Bransford who reported intervertebral fractures in more than 53% of cervical fracture cases (Type I: 53% and a portion of Type III cases) [23]. 

This difference in fracture location may stem from the different ossification patterns observed between cervical and thoracolumbar DISH. The DISH lesion was described by Resnick as flowing calcification along the anterolateral aspect of the thoracic vertebrae, with thick bridging ossifications spanning the intervertebral space [2]. The anterolateral mass in the thoracic spine is observed mainly on the right side of the vertebral column, since the pulsating aorta acts as a mechanical barrier for soft tissue calcification; left-sided hyperostosis was observed in patients with viscerum inversus that have a right-sided aorta [24]. Similarly, the absence of ossification at the mid-vertebral level can be attributed to the horizontally transversing segmental arteries [1]. Therefore, fractures in the thoracic spine often occur at the vertebral body level where the hyperostosis is minimal. Without the barriers of arteries to affect ossification in the anterior aspect of the cervical spine, the hyperostosis of the cervical spine is observed in the anterior midline and can progress into a uniform bar-like structure that is in contrast to the flowing pattern found in the thoracic spine [25]. Furthermore, the cervical levels that had retained motion the longest often form relatively large bridging ossifications spanning the intervertebral space but have a radiolucent disc extension into the ossified mass [2,26], and this intervertebral area becomes the weak link in the cervical ossified mass that is susceptible to fracture. While the weakest link may be case-specific during the early stages of hyperostosis formation, the morphology of advanced DISH cases is sufficiently different to offer a possible explanation for the difference in fracture position between cervical and thoracolumbar cases.

### 4.3. Neurological Status and Course

Minor trauma was the cause of injury in half of the cases in both groups, but the neurological status was significantly more severe in the cervical fracture group, with 76.2% sustaining a SCI, while 65.7% of thoracolumbar fracture cases exhibited no initial neurological symptoms. The structural vulnerability of the cervical spine compared to the semi-rigid construction of the thoracolumbar spine may be partly responsible for this difference. Another factor may be the higher cord-canal ratio of the cervical spine, coupled with the presence of concomitant OPLL which would compress the spinal cord. The prevalence of OPLL in 47.0% of cervical fracture cases was in line with the report from Soraya who found 48.7% of OPLL cases examined by whole-spine CT to have concomitant DISH [27]. 

A striking revelation of this study was the high incidence of neurological deterioration in the thoracolumbar spine fracture group. This group of thoracolumbar cases that experienced neurological deterioration started out with only seven out of 47 patients suffering SCI (14.9%), but ended up with 35 out of 47 patients suffering from permanent symptoms of SCI (74.5%). More than half of these cases suffered from doctor’s delay, with nine cases initially diagnosed as compression fractures and three cases as contusions. Considering that pain was reported to be mild in 12 cases and that seven cases had no apparent trauma episode, it is understandable that the treating physicians did not immediately appreciate the risk of neurological deterioration and arrived at their diagnosis relying only on plain radiographs. The fact that even a minor fracture in a DISH-ankylosed spine can progress into a three-column unstable fracture that is at high risk for SCI is now general knowledge in the spine surgeon community, but this knowledge is not sufficiently shared by primary care and emergency room physicians. Once plain radiology establishes that the patient’s spine is ankylosed by DISH, it is highly recommended that the patient be evaluated by CT or MRI to rule out fractures, especially when associated with any trauma [28].

Furthermore, patients should also be well informed of the risk of fractures in the DISH-affected spine. Of the 44 cases in this study with patient’s delay, 13 cases (30.0%) suffered delayed neurological deterioration because they did not visit a hospital immediately after injury. Considering that DISH affects the elderly population, the patients should be educated on the risks of DISH-associated fractures as soon as DISH is diagnosed, and then reinforced during follow-up visits to the clinic. DISH patients should also be told that fractures may not be immediately apparent especially when examined only by plain radiographs, and that they should immediately seek medical attention if they suffer from any neurological deterioration later on. 

### 4.4. Mortality

The six-month mortality was significantly higher in the cervical group at 15.5% compared to the 5.0% mortality in the thoracic group. Mortality was significantly higher in SCI cases of the cervical group and the combined study group compared to non-SCI cases, but no apparent difference was observed between cases with and without SCI in the thoracic group. The mortality rates of DISH-associated spinal fractures reported in the literature are widely variable, with Westerveld reporting a three-month mortality of 20.0% in surgically treated DISH patients [10] and Caron reporting a one-year mortality of 32% in patients with fractures of ankylosing spinal disorders [9]. The authors recommended surgical treatment because conservative treatment had a higher mortality rate (51%) compared to surgery (23%). On the other hand, Schoenfeld reported that the mortality of DISH patients suffering from spinal fractures was not higher compared to non-ankylosed control patients [16]. 

## 5. Conclusions

The results of the present study illustrate that fractures in the DISH-ankylosed spine can cause widely diverging clinical sequelae according to fracture level. Cervical fracture cases suffer from a higher degree of SCI and mortality, partially stemming from the higher cord-canal ratio in the cervical spine and the preexisting spinal cord compression caused by associated OPLL. Compared to cervical fracture cases, the initial condition of thoracolumbar fracture cases is considerably less severe. However, the mild symptoms in thoracolumbar fracture cases can lead to a false sense of security both for the patient and physician, along with a severe sense of despair if neurological deterioration occurs. The authors hope that the findings of the present study will lead to increased vigilance in the assessment of DISH trauma cases and a decrease in DISH patients suffering from delayed neurological deterioration.

## Figures and Tables

**Figure 1 jcm-09-00208-f001:**
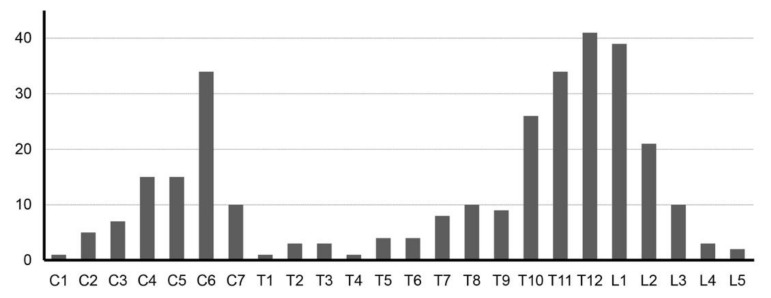
Fracture distribution in diffuse idiopathic hyperostosis (DISH) spines. Distribution of fracture sites in DISH-ankylosed spines, including all multiple fracture sites in the 25 cases that suffered more than one fractured vertebra, demonstrating a bimodal distribution of fractures with peaks at the lower cervical level and the thoracolumbar junction. (C: cervical, T: thoracic, L: lumbar).

**Figure 2 jcm-09-00208-f002:**
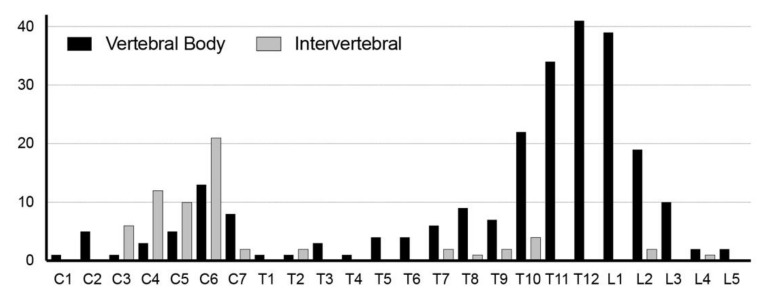
Vertebral fracture pattern. Fracture pattern according to vertebral level, revealing that cervical fractures had more fractures at the intervertebral disc level, while thoracolumbar fractures occurred mainly at the vertebral body.

**Figure 3 jcm-09-00208-f003:**
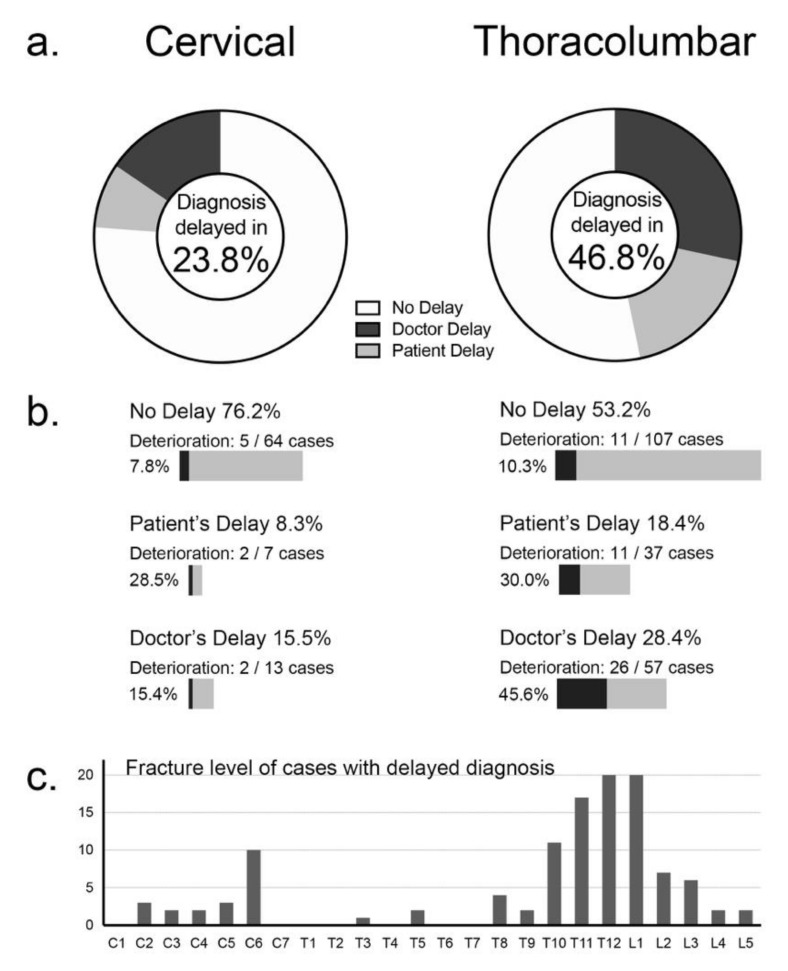
Delay in fracture diagnosis and its effect on neurological deterioration. The percentage of fracture cases with a delay in diagnosis, either due to patient- or doctor-related factors were significantly higher in the thoracolumbar group (**a**). Neurological worsening was significantly higher in the thoracolumbar group and in cases with delayed diagnosis (**b**). The distribution of cases with delayed diagnosis was similar to fracture distribution (**c**). SCI: spinal cord injury.

**Figure 4 jcm-09-00208-f004:**
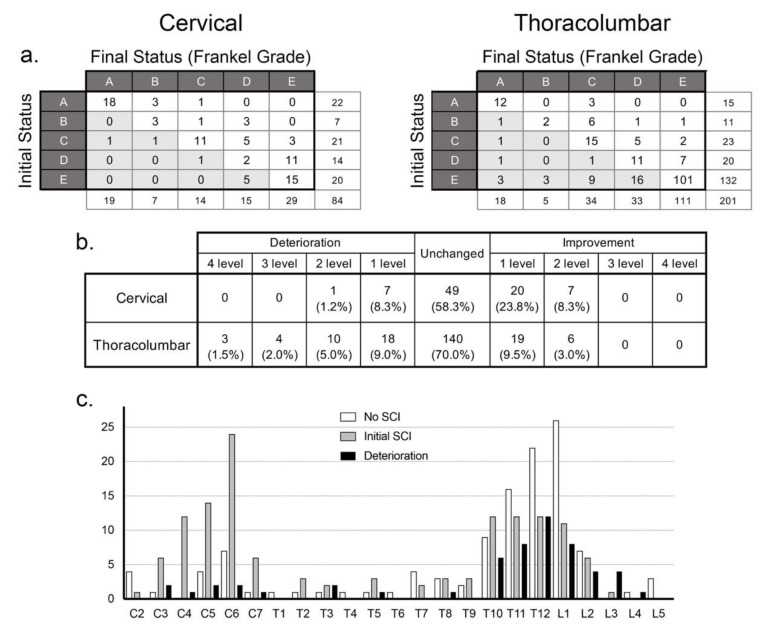
Neurological course of cervical and thoracolumbar fracture cases. The initial and final neurological status of the cervical and thoracolumbar fracture groups according to Frankel grade (**a**) and the change in Frankel grade (**b**) reveal that while cervical cases had a higher ratio of spinal cord injury, thoracolumbar cases experienced higher rates of neurological deterioration, especially in cases with fractures in the thoracolumbar junction area (**c**).

**Figure 5 jcm-09-00208-f005:**
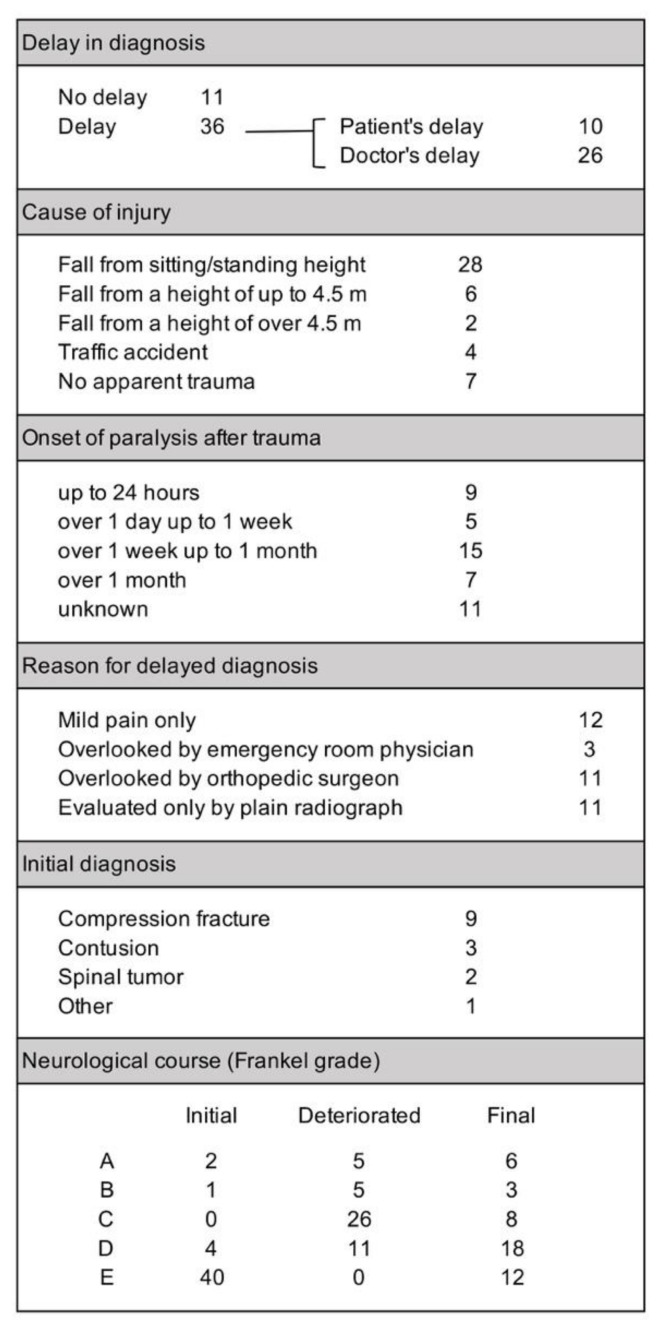
Characteristics of thoracolumbar fracture cases that experienced neurological deterioration.

**Table 1 jcm-09-00208-t001:** Demographics and causes of injury.

	Cervical	Thoracolumbar	*p* value
Cases	84	201	
Gender			
Male	77 (91.7%)	144 (71.6%)	<0.01 *
Female	7 (8.3%)	57 (28.4)	
Age (years, mean ± std)	74.5 ± 9.0	75.4 ± 9.8	0.434
Body Mass Index (kg/m^2^, mean ± std)	22.9 ± 3.9	23.7 ± 3.7	0.181
Past Medical History			
Diabetes	26 (31.0%)	41 (20.4%)	0.055
Kidney disease	2 (2.4%)	11 (5.5%)	0.254
Liver disease	0	5 (2.5%)	—
Osteoporosis	1 (1.2%)	4 (2.0%)	0.64
History of past fractures	0	2 (1.0%)	—
Malignancies	4 (4.8%)	21 (10.4%)	0.122
Cause of Injury			<0.05 *
Fall from standing/sitting position	48 (57.1%)	100 (49.8%)	
Fall from a height	22 (26.2%)	59 (29.4%)	
Traffic accident	13 (15.5%)	21 (10.4%)	
No apparent trauma	1 (1.2%)	21 (10.4%)	<0.0125 *

*: statistically significant difference.

**Table 2 jcm-09-00208-t002:** Accompanying ligament ossification and ankylosing of posterior elements at the site of fracture.

	Cervical	Thoracolumbar	*p* value
Ossification at fracture site			
Ossification of posterior longitudinal ligament (OPLL)	39 (47.0%)	5 (2.5%)	<0.01 *
Ossification of ligamentum flavum (OLF)	1 (1.2%)	1 (0.5%)	0.524
Posterior elements	19 (22.6%)	123 (62.1%)	<0.01 *
Fracture site			
Anterior	84 (100%)	201 (100%)	
Intervertebral	55 (65.5%)	49 (24.4%)	<0.01 *
Vertebral body	29 (34.5%)	152 (75.6%)	<0.01 *
Posterior	39 (46.4%)	134 (67.7%)	<0.05 *
Fracture pattern (Caron classification)			
Type 1 (disc injury)	31	17	<0.125 *
Type 2 (body injury)	25	134	<0.125 *
Type 3 (anterior body or posterior disc injury)	14	21	
Type 4 (anterior disc or posterior body injury)	9	16	

**Table 3 jcm-09-00208-t003:** Treatment, complications, and mortality of DISH-associated fracture cases.

	Cervical	Thoracolumbar	*p* value
Conservative treatment	9 (10.7%)	36 (17.9%)	0.129
Surgical treatment	75 (89.3%)	165 (82.1%)
Anterior fusion	1 (1.2%)	0	0.177
Posterior fusion	67 (79.8%)	153 (76.1%)
Anteroposterior fusion	7 (8.3%)	12 (6.0%)
Complications	31 (36.9%)	55 (27.4%)	0.11
representative conditions:	respiratory: 13	instrument-related: 8	
	UTI: 4	UTI: 8	
	dysphagia: 3	respiratory: 6	
	wound infection: 2	wound infection: 6	
	DVT: 2	DVT: 3	
Death (within 6 months)	13 (15.5%)	10 (5.0%)	<0.05 *
causes:	pneumonia: 4	pneumonia: 2	
	heart failure: 1	heart failure: 1	
	renal failure: 1	septic cholecystitis: 1

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
