# Peer review of "A Comparison of Cervical and Thoracolumbar Fractures Associated with Diffuse Idiopathic Skeletal Hyperostosis—A Nationwide Multicenter Study"

_jcm, 2020, doi:10.3390/jcm9010208_

Round 1
Reviewer 1 Report
The manuscript is an interesting multicenter retrospective review of cervical(CE) and thoracolumbar(TL) fractures associated with diffuse idiopathic skeletal hyperostosis(DISH) aimed to assess and compare the timing of diagnosis, neurological status and radiological characteristics of injuries. 84 CE cases and 201 TL lesions were selected and included in the study. Demographic characteristics, site of the fracture, CT assessment, etc were recorded. The authors concluded that Cervical fracture cases suffer from a higher degree of spinal cord injury and mortality, partially stemming from the higher cord-canal ratio in the cervical spine and the preexisting spinal cord compression.
The article is very interesting but several are controversial issues:
Comment 1:Please report the aim of the study in the abstract.
Comment 2: In the Abstract, Methods are not clearly described.
Comment 3: Please include the diagnostic algorithm in the introduction.
Comment 4: In Introduction report the mains treatments of spine facture in DISH affected-patients.
Comment 5: Please report the aim of the study (line 84-86) at the bottom of the introduction section and summerize the period between lines 86 and 92.
Comment 6: the phrases from line 92 at 96 should be reported in methods.
Comment 7: The Methods is confusing, please organize the section with some subsections, as well as, in results.
Comment 8: Please report center inclusion criteria.
Comment 9: It could be interesting for the reader to know the correlation between the Ossification at the fracture site and age or the fracture site and cause of injury. Please add
Comment 10: Report the main findings at the top of the section.
Comment 11: The first part of the discussion should focus more on the results.
Comment 12: line 312, please report the extended name of AS
Author Response
We thank Reviewer 1 for the constructive comments that have been invaluable in improving our manuscript. We have attached a file in which we have responded to the reviewer's comments in a point-by-point fashion.

Reviewer 2 Report
The manuscript “A comparison of cervical and thoracolumbar fractures 2 associated with diffuse idiopathic skeletal 3 hyperostosis - a nationwide multicenter study” by Hiroyuki Katoh et al. aimed to evaluate the to examine the differences between fractures of the cervical and 37 thoracolumbar ankylosed spine in 285 cases with DISH. The conclusion was that cervical fracture cases suffer from a 437 higher degree of spinal cord injury and mortality; on contrary the initial condition of thoracolumbar fracture cases is considerably less severe, but the mild symptoms in thoracolumbar fracture cases can lead to a false sense of security both for the patient and physician, along with a severe sense of despair if neurological deterioration occurs.
COMMENTS
1. The Authors should describe in more detail the fracture type (vertebral body fracture versus intervertebral fractures).
2. The characteristics of the study population in addition to diabetes should be reported (kidney or liver diseases, previous fractures etc )
3. How many people suffered by osteoporosis. In particular the should reported if in the subjects were BMD by DXA were assessment.
4. As the population is elderly, the fractures could be on an osteoporotic basis
5. The “result” section seems to be excessively long and should be shortened, in particular where the Authors repeat the table or figure results.
6. English grammar and syntax should be improved.
Author Response
We appreciate the time that Reviewer 2 took to review our manuscript. By addressing the comments from Reviewer 2, we believe that our manuscript has been significantly improved. We hope that the revised text achieves the quality required for publication in JCM.

Round 2
Reviewer 1 Report
The article is now suitable for the publication